

# Phylogeographic structure in three North American tent caterpillar species (Lepidoptera: Lasiocampidae): *Malacosoma americana, M. californica,* and *M. disstria*

Linda A. Lait and Paul D.N. Hebert

Centre for Biodiversity Genomics, University of Guelph, Guelph, Ontario, Canada

## ABSTRACT

While phylogeographic structure has been examined in many North American vertebrate species, insects have received much less attention despite their central ecological roles. The moth genus *Malacosoma* (Hübner, 1820), is an important group of forestry pests responsible for large-scale defoliation across much of the Nearctic and Palearctic. The present study uses sequence variation in the mitochondrial cytochrome *c* oxidase 1 (COI) gene to examine the population genetic structure of the three widespread *Malacosoma* species (*M. americana, M. californica,* and *M. disstria*). Populations of all three species showed highest diversity in the south, suggesting that modern populations derived from southern refugia with loss of variation as these lineages dispersed northwards. However, despite similar life histories and dispersal abilities, the extent of regional variation varied among the taxa. *M. americana,* a species restricted to eastern North America, showed much less genetic structure than the western *M. californica* or the widespread *M. disstria*. The regional differentiation in the latter reflects the likely derivation of modern lineages from several refugia, as well as taxonomic uncertainty in *M. californica*. In these respects, the three species of *Malacosoma* share phylogeographic patterns similar to those detected in vertebrates which are characterised by greater phylogeographic breaks in the western half of the continent and limited structure in the east.

## INTRODUCTION

The patterns of genetic variation in species and the processes which underlie them are of particular interest to evolutionary biologists. Diverse factors, both historical and contemporary, influence how variation is distributed among populations; these include geological and climatic events, and the presence of physical and behavioural barriers (*Avise, 2004*). Past glaciations have had a major impact on the extent and patterning of genetic structure in Northern Hemisphere species (*Hewitt, 2000*). In North America, ice sheets covered much of present-day Canada and the northern United States, temperatures in

Corresponding author
Linda A. Lait, llait@uoguelph.ca

ice-free areas were cooler than today, and sea levels dropped by up to 140 m (*Pielou, 1991*; *Barendregt & Irving, 1998*; *Dyke et al., 2002*). The distributions of many species were fragmented with their populations persisting in small ice-free refugia (*Hewitt, 1996*; *Hewitt, 2000*; *Stewart & Lister, 2001*). In addition, physical and ecological barriers influenced genetic structure by their impacts on dispersal and gene flow. Recent environmental changes, both anthropogenic and natural, are now causing range shifts and population changes with varied impacts on both inter- and intraspecific genetic variation (*Walther et al., 2002*; *Chen et al., 2011*).

Comparisons of genetic variation spanning multiple species can identify both general and species-specific patterns, revealing how particular life history characteristics impact population structure. Species-specific traits, such as dispersal ability or niche requirements, may affect how a species responds to environmental, climatic, and geological changes. For example, the majority of northern species experienced major population declines reflecting the loss of habitat during the Pleistocene glaciations, while the interglacials favoured range expansion and population growth (*Nilsson, 1983*; *Pielou, 1991*; *Hewitt, 2000*; *Hewitt, 2004*). In contrast, cold-adapted species experienced habitat loss, often retreating to high altitude and high latitude locations during interglacials (*Stewart & Lister, 2001*; *Dalén et al., 2005*; *Galbreath et al., 2009*; *Stewart et al., 2009*). Topographic features also have differing effects, with mountains and rivers restricting gene flow in some species while acting as dispersal corridors for others. For example, the Rocky Mountains prevent gene flow between some populations (*Crease et al., 1997*; *Burg et al., 2005*), but provide habitat as ''sky islands'' with the intervening lowlands limiting gene flow in others (*Knowles, 2000*; *DeChaine & Martin, 2005*; *Galbreath et al., 2009*).

Past studies of phylogeographic structure in terrestrial organisms have largely examined vertebrates. Given their high diversity and abundance, phylogeographic patterns in insects have been understudied; past work has revealed diverse outcomes ranging from global panmixis (*Alvial et al., 2007*; *Correa et al., 2017*) to highly fragmented, structured populations (*Dinca, Dapporto & Vila, 2011*; *Frantine-Silva et al., 2017*; *Karthika et al., 2017*). Phylogeographic studies of Lepidoptera have employed both nuclear and mitochondrial markers, particularly the cytochrome *c* oxidase 1 (COI) locus (*Vandewoestijne et al., 2004*; *Craft et al., 2010*; *Kirichenko et al., 2017*). This study represents the first step in a broad investigation of phylogeographic patterns in North American Lepidoptera.

*Malacosoma* (Hübner, 1820) is a Holarctic genus found across much of North America, Europe, and Asia, with six species native to Canada and the United States (*Franclemont, 1973*). While males are strong fliers, females are usually sedentary until they deposit their egg mass (*Stehr & Cook, 1968*; *Franclemont, 1973*), suggesting that the dispersal of maternal markers will be low. As their name (tent caterpillar) suggests, they build large tents or moulting mats which can accommodate a large number of larvae (*Franclemont, 1973*). All six species feed on diverse deciduous trees and shrubs, with host preferences varying by taxon and region (*Stehr & Cook, 1968*; *Franclemont, 1973*; *Parry & Goyer, 2004*). The group contains a number of important forestry pests, species that experience cyclical outbreaks which often lead to extensive forest defoliation (*Hildahl & Reeks, 1960*; *Stehr & Cook, 1968*;

*Roland, 1993*). Despite these impacts, there have been few studies of population genetic structure in these moths. One study, which assessed allozyme variation in *Malacosoma americana* (Fabricius, 1793) from eastern United States, found limited variation and a lack of regional genetic differentiation (*Costa & Ross, 1994*). A second employed microsatellites and short DNA sequences to compare five populations of *Malacosoma californica pluviale* in coastal British Columbia (*Franklin, Myers & Cory, 2014*). Although high levels of variation were evident, there was little genetic differentiation between island and mainland populations. Both studies found limited genetic differentiation at a relatively small geographic scale.

This study examines the population genetic structure of the three widely distributed North American *Malacosoma* species, the eastern tent caterpillar *M. americana*, the western tent caterpillar *M. californica* (Packard, 1864), and the forest tent caterpillar *M. disstria* (Hübner, 1820) (Fig. 1). We employ the 659 bp cytochrome *c* oxidase I gene region at the continental scale to determine whether limited dispersal abilities and contemporary barriers are preventing movement in these species, what role the Pleistocene glaciations may have played in their current structure and distribution, and if these three *Malacosoma* species show concordant phylogeographic patterns or if there are differences in the patterns observed that may be explained by their life history characteristics. By understanding what influences have impacted the population genetic structure in these species we can add to our understanding of phylogeography in North American insects and we can begin to predict how the populations may expand with changing climate conditions.

## MATERIALS AND METHODS

### Phylogenetic analyses

Sequences of the 658 bp barcoding region of cytochrome *c* oxidase I (COI) for all *Malacosoma* specimens from the United States and Canada were downloaded from the Barcode of Life Data System (BOLD; see Table S1; *Ratnasingham & Hebert, 2007*). Locations were recorded by state or province. Sequences were aligned in MEGA v6 (*Tamura et al., 2013*). In order to verify that the samples examined belonged to monophyletic species, a Bayesian network was constructed in BEAST v2.3 (*Bouckaert et al., 2014*) using the Hasegawa, Kishino, and Yano model with gamma-distributed rate variation and allowing for invariable sites (HKY $+ \Gamma + I$). The analysis was run for 10,000,000 MCMC steps, and sampled every 2,000 steps with a 25% burn-in. The lasiocampid *Phyllodesma americana* (GenBank accession number JF842281) was used as the outgroup.

### Genetic analysis

Intraspecific analyses were performed on the three widespread *Malacosoma* species: *M. americana*, *M. californica*, and *M. disstria*. These species were selected based on distribution and available sample size (Table 1, Fig. 1). Haplotypes were assigned to each species with TCS v1.21 (*Clement, Posada & Crandall, 2000*) and confirmed manually. Genetic diversity was measured with haplotype ($H_d$) and nucleotide ($\pi$) diversity indices, calculated in DNAsp v5.10 (*Rozas et al., 2003*; *Librado & Rozas, 2009*). In order to test for population genetic structure an analysis of molecular variance (AMOVA; *Excoffier,*

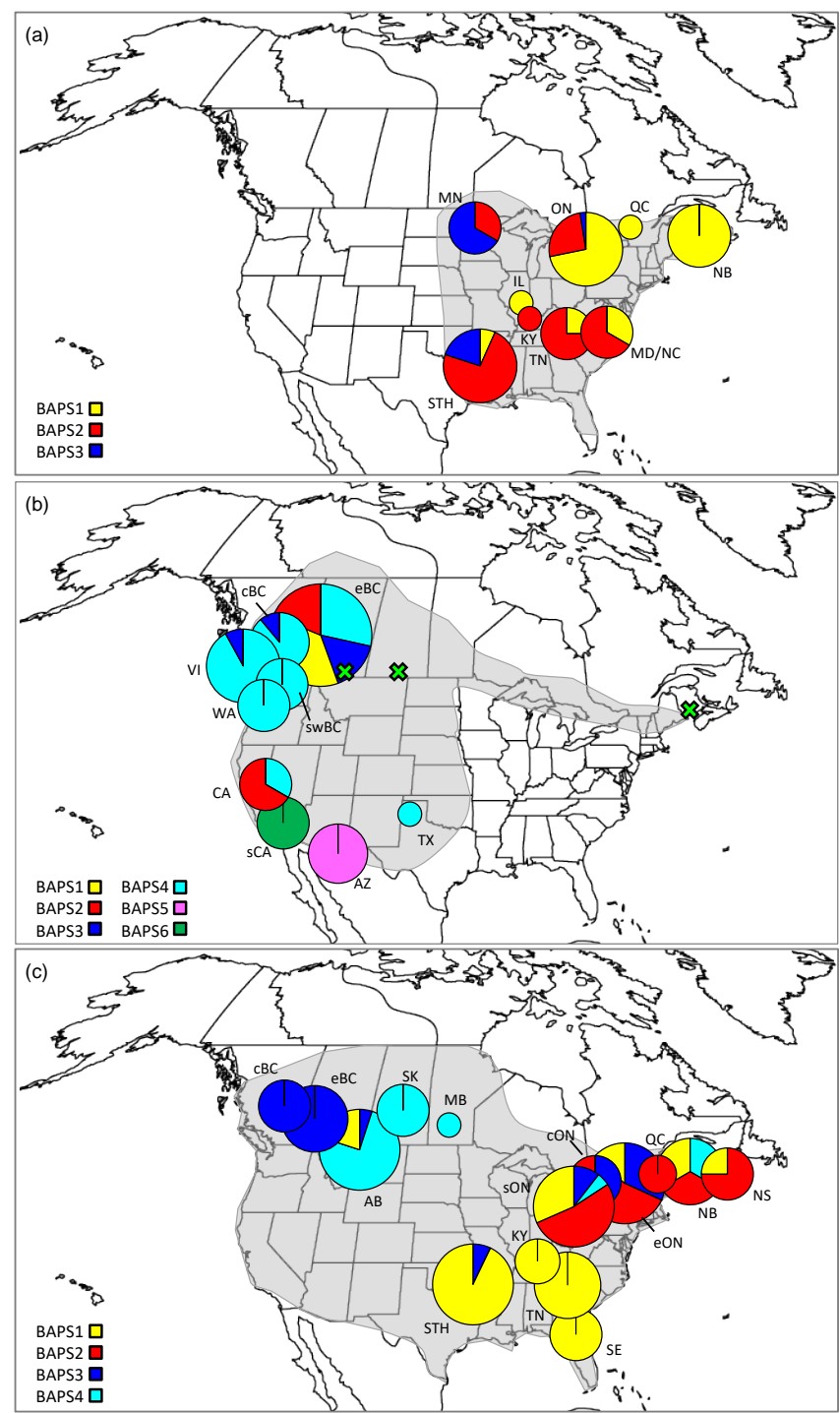

**Figure 1 Distributions, sampling locations, and Bayesian cluster membership for three *Malacosoma* species.** Approximate distributions (shaded) and sampling locations for (A) *Malacosoma americana*, (B) *M. californica*, and (C) *M. disstria*. The pie charts represent the distribution of BAPS groups, scaled for sample size. The green crosses represent the omitted AB, SK, (continued on next page...)

**Figure 1 (…continued)**
and NB samples. Sampling locations are as follows: Alberta (AB), Arizona (AZ), British Columbia (BC; central [c], eastern [e], and southwest [sw]), California (CA; southern [s]), Kentucky (KY), Maryland (MD), Minnesota (MN), New Brunswick (NB), North Carolina (NC), Nova Scotia (NS), Ontario (ON; central [c], eastern [e], and southern [s]), Saskatchewan (SK), Tennessee (TN), Texas, (TX), Vancouver Island BC (VI), and Washington (WA). The STH includes Arkansas (AR), Oklahoma (OK), and TX combined, and SE includes Florida (FL), Georgia (GA), and NC combined.

**Table 1** **Sample details for three *Malacosoma* species.** Sample size ($n$), number of sampling locations (loc), variable sites (VS), mean % pairwise distances (PD), number of haplotypes ($h$), overall haplotype ($H_d$) and nucleotide ($\pi$) diversities, and fixation index ($\Phi_{ST}$) for three *Malacosoma* species. The three $\Phi_{ST}$ values were all highly significant ($p < 0.0001$).

| Species | $n$ | loc | VS | PD $\pm$ SE | $h$ | $H_d$ | $\pi$ | $\Phi_{ST}$ |
|---|---|---|---|---|---|---|---|---|
| *M. americana* | 79 | 12 | 37 | 0.490 $\pm$ 0.303 | 33 | 0.918 | 0.0049 | 0.244 |
| *M. californica* | 207 | 9 | 61 | 0.968 $\pm$ 0.668 | 64 | 0.925 | 0.0097 | 0.477 |
| *M. disstria* | 139 | 19 | 43 | 0.628 $\pm$ 0.383 | 42 | 0.926 | 0.0063 | 0.524 |

*Smouse & Quattro, 1992*) and pairwise genetic differences ($\Phi_{ST}$; 100,000 permutations) were calculated in Arlequin v3.5.1.2 (*Excoffier & Lischer, 2010*). Nearby sampling locations were grouped to increase sample sizes, and a modified false discovery rate was applied to correct for multiple tests (*Benjamini & Yekutieli, 2001*).

To test for the presence of genetic clusters two methods were used: a spatial analysis of molecular variance (SAMOVA) which identifies the maximum between group variance with the use of additional geographic information ($K = 2$ to 6; 1,000 iterations; *Dupanloup, Schneider & Excoffier, 2002*), and a clustering analysis as performed in Bayesian Analysis of Population Structure v5.2 (BAPS; *Corander & Tang, 2007*; *Corander et al., 2008*) which allows the assignment of individuals to genetic clusters with no *a priori* population information. Results from the *Malacosoma* Bayesian analysis were also analysed at the species level. In order to visualise the pattern of variation and relationship among haplotypes, a principal coordinates analysis (PCoA) was run in GenAlEx v6.5 (*Peakall & Smouse, 2006*; *Peakall & Smouse, 2012*), and a statistical parsimony network was constructed in TCS v1.2.1 (*Clement, Posada & Crandall, 2000*).

## RESULTS

### Phylogenetic analyses

A total of 474 *Malacosoma* sequences from five species were downloaded from BOLD. The Bayesian network identified two main lineages within North America: the first group included specimens of *M. constricta* and *M. disstria* with each species forming a well-defined monophyletic group; the second included *M. americana*, *M. incurva*, and *M. californica* (Fig. S1). While *M. americana* was monophyletic, specimens assigned to *M. californica* were paraphyletic, suggesting a species complex. Of particular note, specimens of *M. californica* from Alberta (AB), Saskatchewan (SK), and New Brunswick (NB), as well as those identified as *M. californica pluviale*, did not group with the majority of *M. californica* samples. The taxonomic status of *M. californica* has been widely debated; it is currently viewed as

including six largely allopatric subspecies, many of which have previously been considered distinct species, although Franclemont questioned the validity of *M. californica pluviale* as a subspecies rather than a species (*Franclemont, 1973*). By contrast, both *M. americana* and *M. disstria* have no described subspecies (*Stehr & Cook, 1968*; *Franclemont, 1973*). Further study with additional markers is required to clarify the taxonomic status of lineages within the *M. californica* complex, and to determine how many species should be recognized. As such, the intraspecific analyses here focus only on the large monophyletic portion of *M. californica* and do not include *M. californica pluviale* or the NB, AB, or SK samples (see Fig. 1).

## Cytochrome *c* oxidase I sequences

Intraspecific analyses examined 79 *M. americana* samples, 207 *M. californica* samples, and 139 *M. disstria* samples (Table S1). For each species the 658 bp gene region was highly polymorphic with 37 variable sites defining 33 haplotypes in *M. americana*, 61 variable sites defining 64 haplotypes in *M. californica*, and 43 variable sites defining 42 haplotypes in *M. disstria* (Table 1). There were eight, 13, and 13 anticipated amino acid substitutions, no frameshift mutations, and no stop codons. Fixed nucleotide differences were present in *M. californica* between sCA, AZ, and the other populations of this species, and in *M. disstria* between western (BC, AB, and SK) and eastern groups. There were no fixed differences in *M. americana*. Haplotype and nucleotide diversities were high in all species ($H_d = 0.918$–$0.926$; $\pi = 0.0049$–$0.0097$; Table 1), with diversity generally higher in southern and central populations ($r = -0.284$ to $-0.508$; Table S2, Fig. S2). When populations with small sample sizes ($n < 5$) were removed from the correlation analysis, coefficients were much stronger for *M. americana* and *M. californica* ($r = -0.937$ and $r = -0.703$, respectively), and slightly weaker for *M. disstria* ($r = -0.360$).

## Genetic analyses
### *Malacosoma americana*

Of the three species, *M. americana* showed the least population structure (overall $\Phi_{ST} = 0.24$, $p < 0.0001$), the fewest significant pairwise comparisons (seven out of 15; Table 2), and the lowest diversity (Table 1). The greatest pairwise $\Phi_{ST}$ values were seen between NB and the other populations. The SAMOVA analysis identified four groups ($F_{CT} = 0.263$, $p = 0.02$): MN, NB, and the separation of the remaining samples into northern (ON, MD/NC) and southern (TN, AR/OK/TX) populations, while the Bayesian clustering analysis separated the samples into three clusters (Fig. 1, Table S2): a "northern" group found primarily in ON and NB, and two "southern" groups, one found in all populations except NB, and a smaller group primarily in MN and OK. The distribution of haplotypes in the Bayesian groups was significantly non-random ($X^2 = 47.65$, $p < 0.0001$). The BEAST analysis generally supported the BAPS groups with BAPS1 and BAPS2 forming weakly supported clades while BAPS3 contained the remaining samples (posterior probability = 0.65 and 0.5, respectively; Table S1, Fig. S3). The principal coordinates analysis showed a general separation of the samples into northern and southern groups along coordinate 1 (34.7%; Fig. 2A), while coordinates 2 and 3 explained 11.5% and 8.8% of the variation,

**Table 2 Population pairwise $\Phi_{ST}$ values for three *Malacosoma* species.** Population pairwise $\Phi_{ST}$ values for (a) *M. americana* ($P_{crit} = 0.015$), (b) *M. californica* ($P_{crit} = 0.013$), and (c) *M. disstria* ($P_{crit} = 0.011$). $\Phi_{ST}$ values are given below the diagonal and *p*-values above the diagonal (* $p < 0.05$, ** $p < 0.01$, *** $p < 0.001$). Values significant following correction for multiple tests are shaded. Refer to Fig. 1 for locations.

**(A)**

|  | NB | ON/QC | MN | MD/NC | TN | STH[a] |
|---|---|---|---|---|---|---|
| NB | – | ** | ** | ** | ** | *** |
| ON/QC | 0.195 | – | ** | 0.702 | * | *** |
| MN | 0.637 | 0.395 | – | 0.298 | 0.257 | * |
| MD/NC | 0.481 | 0.000 | 0.326 | – | 0.484 | 0.321 |
| TN | 0.564 | 0.175 | 0.202 | 0.000 | – | 0.376 |
| STH[a] | 0.444 | 0.213 | 0.271 | 0.015 | 0.007 | – |

**(B)**

|  | cBC | eBC | swBC | VI | WA | CA | sCA | AZ |
|---|---|---|---|---|---|---|---|---|
| cBC | – | *** | 0.185 | *** | * | *** | ** | *** |
| eBC | 0.249 | – | ** | *** | * | ** | *** | *** |
| swBC | 0.069 | 0.233 | – | ** | 0.143 | ** | ** | *** |
| VI | 0.403 | 0.313 | 0.247 | – | 0.117 | *** | *** | *** |
| WA | 0.352 | 0.258 | 0.229 | 0.138 | – | * | * | ** |
| CA | 0.448 | 0.241 | 0.422 | 0.641 | 0.500 | – | ** | *** |
| sCA | 0.892 | 0.725 | 0.907 | 0.918 | 0.935 | 0.799 | – | ** |
| AZ | 0.682 | 0.618 | 0.661 | 0.789 | 0.662 | 0.544 | 0.732 | – |

**(C)**

|  | cBC | eBC | AB | SK | cON | sON | eON | NB/NS | STH[a] | TN | SE[b] |
|---|---|---|---|---|---|---|---|---|---|---|---|
| cBC | – | 0.305 | *** | *** | ** | *** | *** | *** | *** | *** | ** |
| eBC | 0.044 | – | *** | *** | *** | *** | *** | *** | *** | *** | *** |
| AB | 0.677 | 0.719 | – | 0.284 | *** | *** | *** | *** | *** | *** | ** |
| SK | 0.948 | 0.978 | 0.026 | – | *** | *** | ** | *** | *** | *** | *** |
| cON | 0.541 | 0.626 | 0.357 | 0.494 | – | 0.319 | 0.897 | * | ** | *** | 0.057 |
| sON | 0.617 | 0.660 | 0.255 | 0.362 | 0.016 | – | 0.264 | 0.109 | *** | *** | * |
| eON | 0.486 | 0.530 | 0.294 | 0.367 | 0.000 | 0.012 | – | * | ** | ** | * |
| NB/NS | 0.769 | 0.808 | 0.284 | 0.468 | 0.201 | 0.037 | 0.134 | – | *** | *** | ** |
| STH[a] | 0.612 | 0.683 | 0.354 | 0.496 | 0.259 | 0.212 | 0.201 | 0.269 | – | *** | 0.241 |
| TN | 0.873 | 0.912 | 0.473 | 0.835 | 0.440 | 0.300 | 0.313 | 0.407 | 0.232 | – | * |
| SE[b] | 0.824 | 0.890 | 0.349 | 0.781 | 0.272 | 0.155 | 0.188 | 0.246 | 0.035 | 0.317 | – |

**Notes.**

[a] STH = AR, OK, TX.
[b] SE = NC, GA, FL.
* $p < 0.05$.
** $p < 0.01$.
*** $p < 0.001$.

respectively. The statistical parsimony network showed little pattern to the variation, with haplotypes generally being closely related, although more divergent haplotypes were present in ON, OK, and MN, and there was a general clustering of southern samples (OK, TX, AR, and TN) and NB samples despite a lack of fixed differences (Fig. 3).

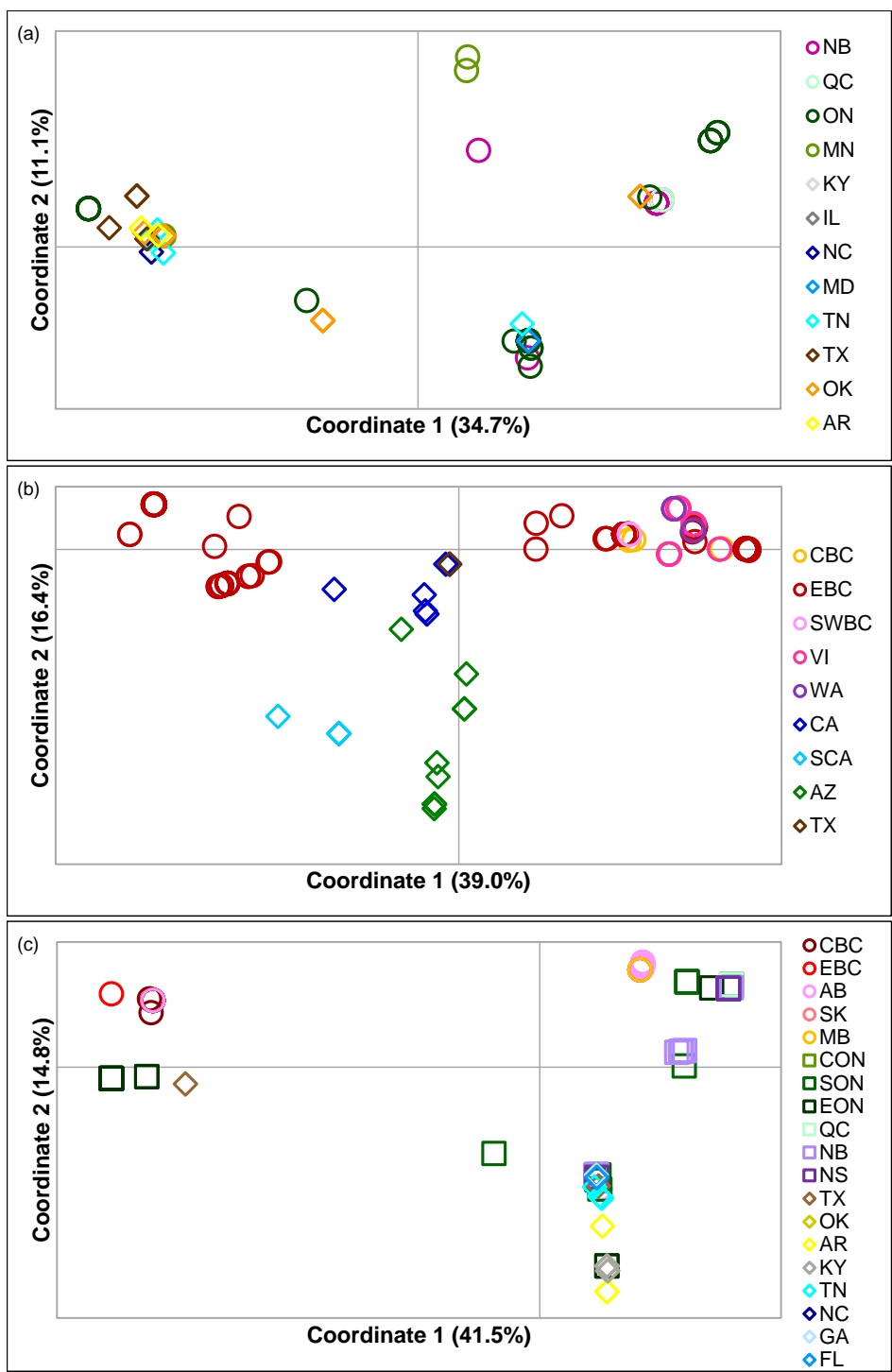

**Figure 2   Principal coordinates analysis for three *Malacosoma* species.** Principal coordinates analysis for (A) *M. americana*, (B) *M. californica*, and (C) *M. disstria*. The northern populations are depicted by a circle or square, while southern populations are represented by a diamond. Samples are colour-coded by sampling location. Refer to Fig. 1 for locations.

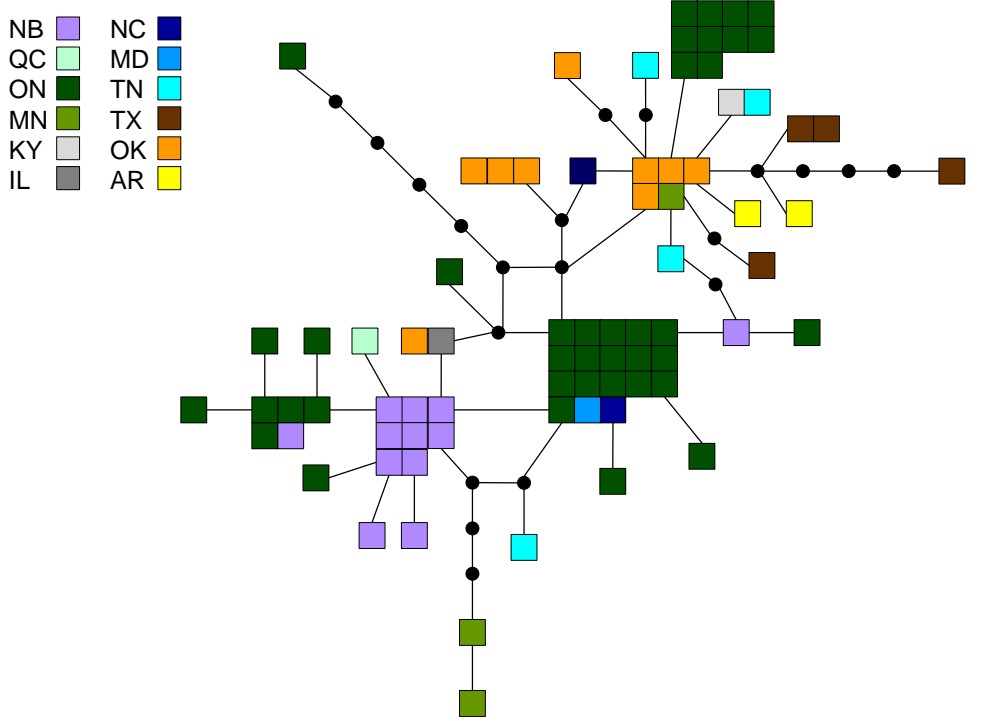

**Figure 3 Statistical parsimony network for *M. americana*.** Statistical parsimony network showing the relationship among the 33 *M. americana* haplotypes. Each square represents one of the 79 sequences colour-coded by location, inferred haplotypes are depicted by black circles, and each line represents a single nucleotide change. Refer to Fig. 1 for locations.

### *Malacosoma californica*

*M. californica* had strong population structure (overall $\Phi_{ST} = 0.48$, $p < 0.0001$), and the highest nucleotide diversity (Table 1). All pairwise comparisons were significant except that between cBC and swBC, and those with WA (22 out of 28; Table 2). These are likely a result of the small WA sample size ($n = 3$), and the small geographic distance between cBC and swBC. The greatest differences existed between sCA and all other populations ($\Phi_{ST} = 0.73–0.94$), and between AZ and all other populations ($\Phi_{ST} = 0.54–0.79$). The SAMOVA analysis identified genetic breaks between three groups: sCA, AZ, and all other populations ($F_{CT} = 0.57$, $p = 0.049$). Bayesian clustering analysis identified six clusters (Fig. 1, Table S2): one in sCA, one in AZ, one in eBC, and three shared between multiple populations. The eBC population had representatives in four clusters. The distribution of samples in the six clusters was highly significant ($X^2 = 486.6$, $p < 0.0001$). When sCA, CA, and AZ were removed from analysis, the distribution was still significantly different than random ($X^2 = 57.02$, $p < 0.0001$). The BEAST analysis identified four main clades with numerous subclades within them: two well-supported clades representing the AZ and sCA samples (posterior probability = 0.95 and 1, respectively), one clade found in eBC with one CA sample (posterior = 0.75), and one found across BC and WA (posterior = 0.77; Fig. S4). The CA samples were found outside of these clades.
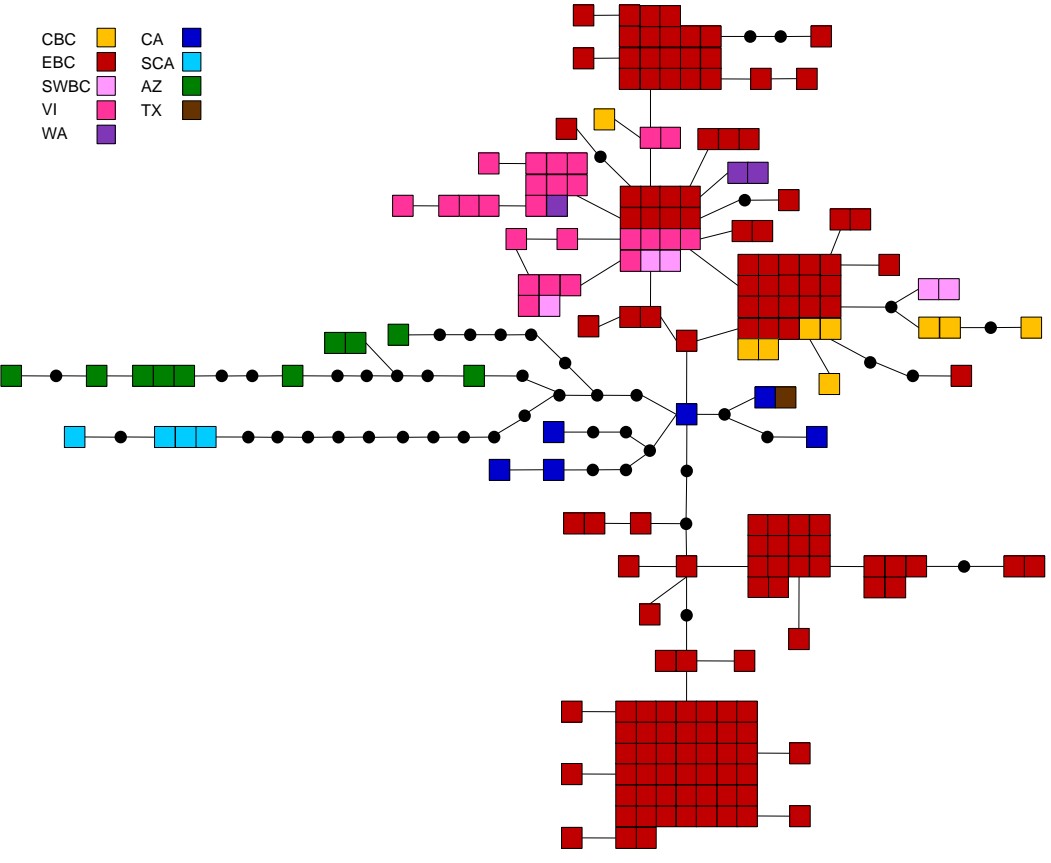

**Figure 4 Statistical parsimony network for *M. californica*.** Statistical parsimony network showing the relationship among the 64 *M. californica* haplotypes. Each square represents one of the 207 sequences colour-coded by location, inferred haplotypes are depicted by black circles, and each line represents a single nucleotide change. Refer to Fig. 1 for locations.

The principal coordinates analysis separated the BC samples into two groups along coordinate 1, while the sCA and AZ samples were separated along coordinate 2 (Fig. 2B). A total of 65.1% of the variation was allocated to the first three coordinates (39.0%, 16.4%, and 9.7%, respectively). The statistical parsimony network showed a similar pattern, separating the AZ, sCA, CA, and TX samples from the more northern populations, with two large groups containing all other samples (Fig. 4). One group consisted of haplotypes from across the Pacific Northwest, and contained all samples from cBC, VI, swBC, and WA, as well as many eBC samples; there were several common haplotypes separated by one to three mutations. The second cluster was restricted to eBC; it was less diverse and contained two common haplotypes, one represented by 44 individuals. Most CA samples had unique haplotypes, while the sCA and AZ haplotypes were divergent with 13–15 (sCA) and 5–16 (AZ) mutations separating them from the nearest population (Fig. 4).

### *Malacosoma disstria*

*M. disstria* had the strongest population structure (overall $\Phi_{ST} = 0.52$, $p < 0.0001$), and 42 of 55 pairwise comparisons were significant following correction for multiple tests (Table 2). The 13 non-significant comparisons all involved ON (with NB/NS, with NC/FL/GA, or among the three ON locations). In contrast, the largest pairwise differences were between BC and all other populations ($\Phi_{ST} = 0.49$–$0.98$). Diversity within populations was generally high with $H_d > 0.7$ in 10 of the 16 populations. The lowest values were in SK ($H_d = 0$) followed by eBC ($H_d = 0.20$; Table S2). A similar pattern was seen with nucleotide diversity with the highest values in the three ON populations ($\pi = 0.0053$–$0.0065$) and TX ($\pi = 0.0137$; Fig. S2).

The SAMOVA analysis identified a genetic break between the BC populations and all other populations (including AB), possibly along the Rocky Mountains ($F_{CT} = 0.52$, $p = 0.015$). Bayesian clustering analysis showed a slightly different pattern with four identified clusters (Fig. 1, Table S2): three were found in the north (two wholly) while one was primarily found in the south. The BC samples fell exclusively in one cluster, the AB,SK, and MB samples were in a cluster together, and the ON, QC, and NB/NS samples generally formed a third cluster. The allocation of samples to clusters was highly significant ($X^2 = 224.7$, $p < 0.0001$). The BEAST analysis identified four main clades, generally supporting the BAPS groups, although with slightly different membership: two well-supported clades found mostly in the west (posterior probability $= 1$ and $0.75$), and two clades present mostly in the east and southeast (posterior $= 0.77$ and $0.39$, respectively; Fig. S5).

The principal coordinates analysis separated the BC populations from most other populations along coordinate 1 (41.5%), and identified a general separation of northern and southern populations along coordinate 2 (14.8%). Coordinate 3 explained 12.6% of the variation (Fig. 2C). The statistical parsimony network identified moderate variation, with five common haplotypes ($n \geq 10$): three found in a single region (BC, AB/SK, or ON), one shared by ON and NB/NS, and one found across several regions in the east and south (Fig. 5). The BC samples formed a separate group (with a single AB sample), while AB, SK, and MB mostly grouped together. The southern samples generally grouped together.

## DISCUSSION

### Population genetic structure

The analysis of mitochondrial COI sequences from three North American *Malacosoma* species showed high levels of variation and diversity, with some highly divergent populations in *M. californica*. All three species show evidence of persistence in one or more southern refugia with subsequent recolonisation of northern regions. Diversity patterns exhibited the characteristic "southern richness, northern purity" (Hewitt, 2004) found across much of the previously glaciated Northern Hemisphere. This was particularly evident in *M. americana* where diversity in southern populations was twice that in northern regions ($\pi = 0.0046$–$0.0061$ versus $0.0013$–$0.0027$), and in *M. californica* where diversity was four-fold higher in Arizona ($\pi = 0.0093$) than in Washington ($\pi = 0.002$; Table S2). In *M. disstria* diversity

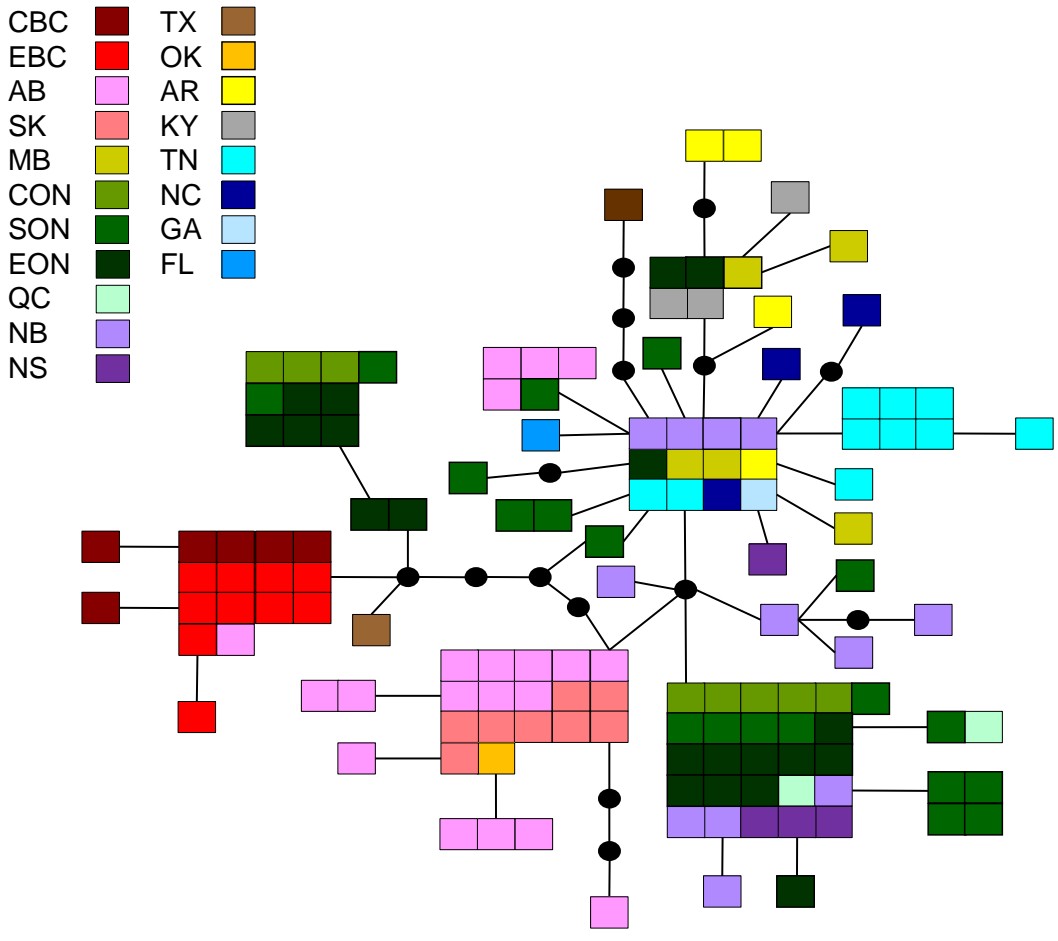

**Figure 5 Statistical parsimony network for *M. disstria*.** Statistical parsimony network showing the relationship among 42 *M. disstria* haplotypes. Each square represents one of the 139 sequences colour-coded by location, inferred haplotypes are depicted by black circles, and each line represents a single nucleotide change. Refer to Fig. 1 for locations.

levels were highest in Texas and in the three Ontario populations, likely representing admixture in the latter. All three species exhibited a negative correlation between latitude and nucleotide diversity (Fig. S2), with the strongest relationships seen in *M. americana* and *M. disstria*.

The two species found in the east, *M. americana* and the eastern portion of *M. disstria*, showed limited population structure consistent with a relatively young evolutionary history and/or high levels of gene flow. This pattern is common in species restricted to eastern North America, and in the eastern portion of the range for more widespread species. For example, many bird species found in the eastern half of North America exhibit limited genetic structure (*Zink, Rootes & Dittmann, 1991*; *Vallianatos, Lougheed & Boag, 2001*; *Veit et al., 2005*), while widespread species show shallow divergence in the east (*Klein & Brown, 1994*; *Graham & Burg, 2012*; *Van Els, Cicero & Klicka, 2012*), likely reflecting a single evolutionary origin and extensive contemporary gene flow. Other studies have identified

limited structure in eastern trees (*McLachlan, Clark & Manos, 2005*; *Shaw & Small, 2005*; *Gerardi et al., 2010*) and mammals (*Petersen & Stewart, 2006*), with the exception of more distinct southern populations (e.g., Texas or Florida). Both *M. americana* and *M. disstria* exhibit this pattern: limited structure in eastern North America with some differences between northern and southern populations. This may be caused by limited ongoing gene flow between regions, likely a result of the limited dispersal capability of female *Malacosoma*. Interestingly, neither species have the genetic break between Atlantic and Gulf coast clades seen in many fish (*Bermingham & Avise, 1986*; *Avise, 1992*), insects (*Vogler & Desalle, 1993*; *Ney & Schul, 2017*), reptiles (*Lamb & Avise, 1992*), birds (*Avise, 1992*), and marine invertebrates (*Herke & Foltz, 2002*; *Young et al., 2002*; see *Soltis et al., 2006* for additional references and an excellent description of this break), suggesting a single southeastern origin.

In addition to limited north-south diversification, *M. disstria* also exhibits an east/west separation (Table 2, Fig. 1), a pattern common in widespread North American species that often reflects multiple evolutionary origins (*Sperling, Raske & Otvos, 1999*; *Gerardi et al., 2010*; *Medina et al., 2010*; *Lait & Burg, 2013*). Lack of strong support for a western or southwestern refugium may be a result of the paucity of samples in this region; the presence of a western refugium may be suggested by the pattern found in *Populus tremuloides*, the favoured host of *M. disstria*, that shows evidence of two genetic clusters, one in the southwest and one in the north and east, with higher diversity in the southwest group (*Callahan et al., 2013*). Many other continent-wide species also possess a large group with low diversity across northern and eastern areas, and multiple diverse groups west of the Rocky Mountains (*Ball & Avise, 1992*; *Byun, Koop & Reimchen, 1997*; *Graham & Burg, 2012*; *Van Els, Cicero & Klicka, 2012*), a pattern linked to a single refugium for the east and multiple isolated refugia in the west. The high genetic diversity in the three Ontario populations (Table S2, Fig. S2), as well as the presence of all four Bayesian clusters meeting in these populations (Fig. 1), supports the possibility of secondary admixture in this central region. Additional sampling between the Ontario populations and the western populations, as well as from the southwestern portion of the range, should help to clarify whether the mixing seen here is indicative of recolonisation from multiple refugia, or if it suggests a diverse source population.

The western *M. californica* possessed a very different pattern of variation with strong population genetic structure and very distinct populations (AZ and sCA) separated by multiple fixed differences which may represent different subspecies or ecotypes. This pattern is common in many southwestern and western species, and is explained by both the current and historical topography of western North America: four major mountain ranges (Cascade, Coastal, Rocky, and Sierra) run along a north-south axis, and large plains and deserts abound, all of which contribute to a complex habitat mosaic. This heterogeneity, coupled with the resulting complex glacial histories of the region, has resulted in extensive structuring in many birds (*Barrowclough et al., 2004*; *Lait et al., 2012*; *Van Els, Cicero & Klicka, 2012*), mammals (*Byun, Koop & Reimchen, 1997*; *Riddle, Hafner & Alexander, 2000*; *Galbreath et al., 2009*), insects (*Brown et al., 1997*), and plants (*Golden & Bain, 2000*; *Richardson, Brunsfeld & Klopfenstein, 2002*; *Johansen & Latta, 2003*). Given the divergence

of the AZ and sCA populations (0.9–1.9% divergence from the nearest population, CA), it is likely that they have been isolated for multiple glacial cycles with limited recent gene flow. The AZ population shows relatively high diversity, suggesting multiple refugia or impassable barriers within this small region. This has been seen in a number of animal (*Orange, Riddle & Nickle, 1999*; *Zink et al., 2001*; *Merrill, Ramberg & Hagedorn, 2005*; *Graham et al., 2013*) and plant (*Frohlich et al., 1999*) species, and is likely due to the particularly heterogeneous nature of Arizona which contains seven ecoregions (*Warshall, 1995*; *Poulos, Taylor & Beaty, 2007*; *Ober & Connolly, 2015*; *Powell & Steidl, 2015*).

**Physical barriers**

Several physical barriers impede gene flow in North American species. In eastern North America, the Appalachian Mountains act as a barrier to movement in plants (*Griffin & Barrett, 2004*; *Joly & Bruneau, 2004*; *Godbout et al., 2005*), reptiles (*Bushar et al., 2014*; *Krysko et al., 2017*), and amphibians (*Church et al., 2003*; *Jones et al., 2006*), while the Mississippi, Tombigbee, and Appalichola Rivers prevent gene flow in many plant and animal species (see *Bermingham & Avise, 1986*; *Avise, 1992*; *Soltis et al., 2006* and references therein). The two *Malacosoma* species in the east do not show genetic breaks along any of these traditional barriers. This may be due to recent colonisation from a single origin or ongoing gene flow in these regions. As the moths can fly, their dispersal capabilities should be greater than that of sedentary plant and reptile species, allowing them to cross rivers. The fact that the Appalachians have not prevented gene flow may indicate the importance of forested valleys as dispersal corridors. All three *Malacosoma* species are generalist herbivores, albeit with host preferences, thus they should encounter suitable habitat more often than strict specialists.

In western North America the main physical barriers are the Rocky, Coastal, Cascade, and Sierra Nevada Mountains (*Crease et al., 1997*; *Nielson, Lohman & Sullivan, 2001*; *Johansen & Latta, 2003*; *Burg et al., 2005*; *Carstens et al., 2005*). The Wyoming Basin and the Great Plains have also been shown to act as dispersal barriers, particularly for species associated with montane, forested, or wetland habitats (*DeChaine & Martin, 2005*; *Wilson et al., 2005*). *Malacosoma disstria* and *M. californica* both exhibit genetic breaks in western regions: *M. disstria* shows a break across the Rocky Mountains (between BC and AB), while *M. californica* has disjunct populations among many of the southwestern deserts (Table 2, Figs. 1 and 2). The Rocky Mountains may also act as a barrier in this species (or species complex); the samples identified as *M. californica* from AB and SK were very different than those in BC (4.9–6.7%), likely representing the described subspecies *M. californica lutescens* as a separate species (the Great Plains tent caterpillar; *Franclemont, 1973*).

## CONCLUSIONS

Despite considerable overlap in their distributions and life history traits, the patterns of variation and levels of population structure in the three *Malacosoma* species varied considerably. The population genetic structure suggests a single origin in the east and a complex evolutionary history in the west. *M. americana*, restricted to the eastern half of the continent, shows limited structure with a north-south trend and greater diversity in

the south. This is consistent with its expansion from a single southern refugium following the last glaciation with limited ongoing gene flow between distance regions. *M. disstria* shows a similar pattern in the east, supporting a single southern refugium, with one or more additional refugia possible in the west. Additional samples are required to elucidate whether the differentiation in its BC population reflects a founder event with subsequent divergence or additional structuring. *M. californica* shows the greatest degree of structure and differentiation, consistent with multiple evolutionary origins in the west and southwest. This study shows the utility of existing DNA barcodes in identifying patterns of genetic structure in insect species, which can uncover previously unknown evolutionary histories and suggest further avenues to explore.

## ACKNOWLEDGEMENTS

We would like to acknowledge all of the contributors to the Barcode of Life Database including the collections, laboratory, and bioinformatics staff at the Centre for Biodiversity Genomics, and researchers from around the globe.

### Funding
This work was supported by a Natural Sciences and Engineering Research Council (NSERC) Discovery Grant to Paul D.N. Hebert and is a contribution to the "Food From Thought" research program funded by the Canada First Research Excellence Fund. The funders had no role in study design, data collection and analysis, decision to publish, or preparation of the manuscript.

### Grant Disclosures
The following grant information was disclosed by the authors:
Natural Sciences and Engineering Research Council (NSERC).
Canada First Research Excellence Fund.

### Competing Interests
The authors declare there are no competing interests.

### Author Contributions
- Linda A. Lait conceived and designed the experiments, analyzed the data, prepared figures and/or tables, authored or reviewed drafts of the paper, approved the final draft.
- Paul D.N. Hebert conceived and designed the experiments, authored or reviewed drafts of the paper, approved the final draft.

### Data Availability
The research did not generate any new sequences. All sequences were already available in the Barcode of Life Database. A list of which sequences were used is available in Table S1.

## Supplemental Information

Supplemental information for this article can be found online at http://dx.doi.org/10.7717/peerj.4479#supplemental-information.

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
