# Peer review of "Phylogeographic structure in three North American tent caterpillar species (Lepidoptera: Lasiocampidae): Malacosoma americana, M. californica, and M. disstria"

_PeerJ, doi:10.7717/peerj.4479_

## Round 0.1 · original submission · Major Revisions

I have now received the opinions of two reviewers regarding your manuscript. In my opinion, generally, the study is interesting and brings new data to light. Unfortunately, the manuscript still needs significant improvement before it can be considered for publication.

Both reviewers point out important outstanding issues with the manuscript as it currently stands, importantly:

1. I agree with both reviewers in that the manuscript lacks cohesion and clarity in certain aspects. I suggest further clarification of the principal objective, followed by a clear and concise explanation of why each methodological analysis was carried out. The analytical part is a tour de force: there is a descriptive genetic analysis, a splittling analysis, a statistical parsimony network, a bayesian network, a PCA, a cluster analyses, several AMOVAs, a phylogenetic tree which is not detailed in methods (appears in results, should be in methods). All these analysis need to be explained, justified and included in the methods section in an objective, logical order. Ideally these explanations should be followed by what the expectation is for a given result.

2. Once the evidence is clearly laid out, I agree with reviewer #2 that the conclusions and inferences need to be better justified. While the authors provide many statistical tests, most are of a descriptive nature and not tests of hypothesis. Thus, the conclusions are not firmly backed by the results (as currently written). I suggest that the authors carefully take in consideration the comments made by reviwer #2, and either provide more support for their conclusions or clarify the level of uncertainty surrounding each particular statement.

3. I also agree with reviewer #2 about the general justification for this piece of research. While it is noteworthy that insects are understudied and that the presented data are important in filling a gap of knowledge, there are certainly other interesting biological aspects that will serve as better justification for the data presented, as well as pitch the article to a wider audience.

·

Basic reporting

The manuscript is well written, with interesting results and discussion. Patterns are well illustrated and sufficient data is available to the reader.

However, there are some important issues concerning the lack of references in some statements (e.g., lines 49-50 and 88-89) and lack of background about the genus (e.g., number of valid species and distribution).

Experimental design

The most serious question that should be considered to enhance the quality of this manuscript is in the structure and coherence between objectives and methods. The objective of the manuscript (lines 98-100) is to assess the genetic structure of three species of Malacosoma and describe its phylogeographic patterns. However, some phylogenetic analyses are described, not just in methods section, but in results as well, although the reasons for such analyses are not clear in the text. After reading the manuscript, it is possible to understand that some previous phylogenetic analyses were necessary to verify the monophyletic groups the authors could work with. Then, some groups were selected, its genetic structure were described and, finally, phylogeographic patterns were discussed. I suggest that a clearer description of this methodological process could be made in methods section.

Despite of that, the patterns found for these three species of Malacosoma contribute greatly to the understanding of the genetic structure of North American species. Considering the lack of studies with insects and other invertebrates, the authors provided an important contribution to the field.

Validity of the findings

Results are well discussed and well illustrated. Although I am not familiar to genetic analyses, it seems that tables provide sufficient information to understand the description of results and discussion.

Additional comments

I made a few notes about the validity of some terms you used. Maybe you should consider reviewing them. Comments about the lack of clarity in methodology and objectives are more important to consider, in my opinion. They are not necessarily wrong, but confuse, which diminishes the quality of your work. Comments were made intending to enhance your manuscript.

·

Basic reporting

Tables and figures too confusing and not well explained

Experimental design

Research questions diffuse and not particularly well articulated

Purpose of particular methods not clear

Validity of the findings

Claims not well supported and mostly based on assumptions rather than robust tests of alternative hypotheses

Additional comments

Review of Lait and Hebert

Overall critique
I had difficulty making sense of the figures and the purpose of the paper. It strikes me that the authors set out to test whether particular historical events influenced the abundance and distribution of genetic variation. For instance, in the discussion beginning with line 295, the authors discuss whether there is evidence of glaciation effects on the distribution of variation, and they introduce some new data and describe an analysis that was not included in the paper (isolation by distance). I have a hard time seeing the pattern they describe in their figures. The statistical parsimony networks are, in my opinion, uninterpretable and the colored circles maps were also difficult to interpret. Additionally, the “genetic breaks” are difficult to see because they are wholly or partially obscured by the circles. Moreover, contrary to their claim, for M. dissitria, there seems to be LESS variation in the south than in the north. My recommendation is to make a specific testable prediction: all three species have lower genetic variation in the north because of the effect of glaciation. It should be relatively straightforward to make a graph with latitude on the x axis and genetic diversity on the y-axis, indicate the southern boundary line of the glacial maxima, and simply assess whether the expected association exists. If it does, that’s interesting and adds more species to the list of species that were greatly impacted by the last glacial maximum. If there is no association, that’s interesting and suggests a number of different scenarios (species not dispersal limited). Finally, isolation-by-distance does not, by itself, “indicate that recolonization happened in stepping-stone fashion.” Overall, I would argue that the claims are not supported (at least by what is presented) and the inferences are weak and overstated.

Further, the authors reported that “The fact that M. americana lacked fixed differences among populations, while M. disstria had only a few between the western and eastern populations, suggests a single refugial origin for both species, likely in the south-eastern United States.” I am unaware that this evidence enables a claim about the number of refugia. It strikes me that the authors would need to establish alternative models that different in the number of refugia and to use some objective criteria for evaluating the fit of the data to each model. Only then would I be convinced that there was a single refuge. (See papers by Steele and Storfer 2006; Waltari et al. 2007; Bao et al. 2015, etc).
The authors did a phylogenetic analysis. It would have been nice to see a full phylogram tree of all haplotypes rather than the un-informative triangles (Figure 2). Also, it is curious why the authors did not include all seven clades in their analysis since each clade appears to be variable. This would have provided more power for estimating the dependence of diversity on latitude.
I was also unclear why the PC plot is so complicated and what, if anything, we are supposed to come away with after trying to figure out what all the colors and symbols mean. Overall, the authors did a poor job of constructing visualizations of data that effectively and efficiently communicate relevant information (in my opinion).
Table 2 is also nightmarish. I would recommend using the data to construct a simple visualization.

I would favor a much more focused story, perhaps one that uses the data to address 1) whether there is a general signal of north-south gradient of genetic variation, and 2) the number and possible location of refugia. These results can then be used to compare with other taxa to assess whether the inferences gain for these species were evident for other species with similar or different biological characteristics.
Overall, I can not recommend acceptance of this manuscript until the authors make the connect between the evidence and claims more robust, or alternatively, emphasize the great amount of uncertainty associated with the claims.
Introduction
The authors begin with the statement… “While phylogeographic structure has been examined in many North American vertebrate species, insects have received much less attention. The present study begins to address this gap by examining population structure in members of the moth genus Malacosoma, an important group of forestry pests.” I am not convinced that the fact that insects “have received much less attention” is sufficient justification for pursuing a study of an insect. Another weak justification for the research: “study represents the first step in a broad investigation of phylogeographic patterns in North American Lepidoptera.” I think it would be much more compelling to justify the work because the biology of the insect can add to our general understanding of phylogeographic structure of temperate organisms.
Methods
I think it would be useful to briefly explain WHY a particular statistical approach is adopted and the parameter that is being estimated.

---

## Round 0.2 · accepted · Accept

Dear authors,

I have now received the 2nd round of reviews from the same reviewers as the 1st round.

While reviewer #2 still has several critiques to the paper, it is my interpretation that the reviewer agrees with the importance and quality of the data being published, but disagrees to some extent with the analytical approach and some of the conclusions. As such, we have reached a point where the reviewer will not be content with any changes that the authors are willing to make, and the authors will not change any more than they already did. In other words, we have reached a point of disagreement as to how the question should be addressed and analyzed - since one other reviewer is perfectly happy with the paper as is, there is an apparent divide in the field.

In my opinion, this is the type of discussion that is enriching to all working in the field, and thus are valuable when made public, where the community has an opportunity to weigh in.

Summarizing, I am confident that the paper is ready for publication, and may even shake up the field a little bit.

Best regards,

dan

# Staff Note: One of the PeerJ Section Editors reviewed this decision and agreed with Dr Lahr's opinion. They did, however strongly suggest that the authors should publish the reviews alongside the paper (as is possible with PeerJ) because: a) the reviews are instructive of the overall process, b) the reviewers – particularly reviewer 2 – makes important points that should make interested readers think more closely about alternative interpretations, and c) both reviewers have agreed to make their names public, indicating that they are also in favor of the reviews being made pubic.

·

Basic reporting

The second version of this manuscript is much better and clearer than the previous one. All the main critics and suggestions made by the reviewers were considered and included by the authors.

Experimental design

Methodology is better described and objectives are well defined.

Validity of the findings

Results are much more clearer than the previous version, with data well described and discussed. In the Conclusion section, the main achievements are highlighted properly and appropriate suggestions for further studies are made.

Additional comments

In my opinion, both authors carefully considered the critics and suggestions made by the reviewers and the editor. The quality of the manuscript was greatly improved and it is, in my opinion, now suitable for publication.

·

Basic reporting

First, I'll begin by congratulating the authors on their revision: they did a good job of addressing all of the criticism . Importantly, they clarified the purpose of the paper. They wrote the work was motivated "...to determine whether limited dispersal abilities and contemporary barriers are preventing movement in these species, what role the Pleistocene glaciations may have played in their current structure and distribution, and if these three Malacosoma species show concordant phylogeographic patterns or if there are differences in the patterns observed that may be explained by their life history characteristics."

I have no problem with the authors publishing this paper. There is good information that needs to be published.

But do I want to be very clear that most of the revision failed to help me understand what the data show, why they do the analyses they do, and whether any hypotheses were tested. For example, one of their goals (stated above) is testing whether their is an effect of life history on patterns of genetic variation. With three species, n = 3. I am certain that there is insufficient power to estimate the effect of life history traits on the distribution of mtDNA sequence variation.

Similarly, the data are DNA sequences from a short section of one gene. I am not sure why they used a multi-locus method (BAPS) for inferring "clusters". I would have preferred a more conventional approach that includes making haplotype trees for each of the three species (33, 64, and 42 "taxa") and perhaps color code taxon labels by latitude or distance from the edge of the ice during the last glacial maximum. I have a hard time see anything in the data visualizations that tell me anything about the role that "Pleistocene glaciations may have played...". In response to my criticism of the largely (and still) confusing statistical parsimony trees, the authors noted that "The full phylogram is not appropriate to include for the purposes I was using it for (to test monophyly). The full figure contains almost 500 samples - this would make it entirely unreadable. and lose the point of the figure." I wasn't asking for ALL sequences to be included, only the unique haplotypes for each species and there are fewer enough the tree will likely be easier to interpret, with appropriate taxon labels, than multi-colored tiles in figures 3-5. Being color blind makes these figures a challenge for me.

The authors remain intent on making very specific claims without evidence of support for alternative hypotheses. For example, "M. americana...shows limited structure with a north-south trend and greater diversity in the south. This is consistent with its expansion from a single southern refugium following the last glaciation with limited ongoing gene flow between distance regions." What other historical events, processes, might generate the same pattern? What does it mean for data to be consistent? Finally, we should lose site of the fact the data are a single gene and so there are no degrees of freedom associated with historical inference. Typically phylogeographic inference necessarily requires MULTIPLE independent loci.

Finally, there were really no changes in the data visualizations to help me better see patterns in the data. The PC graphs have a lot of colors and symbols and I am not sure WHAT the purpose of the graph is relative to the goals of the paper. It does not relate to life history, dispersal ability, or the Pleistocene as far as I can tell.

Overall, while I have no problem with the paper being published, I do think that the inferences based on limited data go far beyond the evidence. What I see is nicely summarized in table 1: three species with different amounts of genetic variation for a bar-code locus.

Experimental design

No comment

Validity of the findings

See above

Additional comments

See above